# iPSCs in Neurodegenerative Disorders: A Unique Platform for Clinical Research and Personalized Medicine

**DOI:** 10.3390/jpm12091485

**Published:** 2022-09-10

**Authors:** Shashank Pandey, Michal Jirásko, Jan Lochman, Alexandr Chvátal, Magdalena Chottova Dvorakova, Radek Kučera

**Affiliations:** 1Department of Pharmacology and Toxicology, Faculty of Medicine in Pilsen, Charles University, Alej Svobody 1655/76, 32300 Pilsen, Czech Republic; 2Biomedical Center, Faculty of Medicine in Pilsen, Charles University, Alej Svobody 1655/76, 32300 Pilsen, Czech Republic; 3Institute of Animal Physiology and Genetics, The Czech Academy of Science, Veveří 97, 60200 Brno, Czech Republic; 4Department of Biochemistry, Faculty of Medicine, Masaryk University, Kamenice 753/5, 62500 Brno, Czech Republic; 5Department of Physiology, Faculty of Medicine in Pilsen, Charles University, Alej Svobody 1655/76, 32300 Pilsen, Czech Republic; 6Department of Immunochemistry Diagnostics, University Hospital Pilsen, 30955 Pilsen, Czech Republic

**Keywords:** induced pluripotent stem cells (iPSCs), Alzheimer’s disease, Parkinson’s disease, diabetic neuropathy, spinal cord injury, personalized medicine

## Abstract

In the past, several animal disease models were developed to study the molecular mechanism of neurological diseases and discover new therapies, but the lack of equivalent animal models has minimized the success rate. A number of critical issues remain unresolved, such as high costs for developing animal models, ethical issues, and lack of resemblance with human disease. Due to poor initial screening and assessment of the molecules, more than 90% of drugs fail during the final step of the human clinical trial. To overcome these limitations, a new approach has been developed based on induced pluripotent stem cells (iPSCs). The discovery of iPSCs has provided a new roadmap for clinical translation research and regeneration therapy. In this article, we discuss the potential role of patient-derived iPSCs in neurological diseases and their contribution to scientific and clinical research for developing disease models and for developing a roadmap for future medicine. The contribution of humaniPSCs in the most common neurodegenerative diseases (e.g., Parkinson’s disease and Alzheimer’s disease, diabetic neuropathy, stroke, and spinal cord injury) were examined and ranked as per their published literature on PUBMED. We have observed that Parkinson’s disease scored highest, followed by Alzheimer’s disease. Furthermore, we also explored recent advancements in the field of personalized medicine, such as the patient-on-a-chip concept, where iPSCs can be grown on 3D matrices inside microfluidic devices to create an in vitro disease model for personalized medicine.

## 1. Introduction

The discovery of induced pluripotent stem cells (iPSCs) technology in 2007 revolutionized pre-clinical research and allowed the development of in vitro disease models for a wide range of disorders, such as neurodegenerative disease, diabetes mellitus, and heart, liver, lung, and kidney disease [1]. To bridge the gap between pre-clinical research and human clinical trial, it is essential to create a more appropriate drug discovery system. The reprogramming of differentiated cell types such as patients’ fibroblast or peripheral blood mononuclear cells (PBMCs) into pluripotent stem cells has been developed and used for drug screening, which has dramatically improved the disease model system for in vitro drug analysis. The application of this technology is used to study neurological diseases such as Alzheimer’s disease (AD), Parkinson’s disease (PD), spinal cord injury (SCI), amyotrophic lateral sclerosis (ALS), multiple sclerosis, and ataxia. It also allows scientists to perform research and understand the effect of newly discovered drugs on complex tissues of human origin, such as brain and cardiac tissues, which are difficult to obtain. In this model, patient-specific iPSCs cell lines are obtained from biopsies or blood cells and maintained. These iPSCs are reprogrammed into specific cell types of interest, thus reiterating the disease condition in vitro in a Petri dish. The proliferation and differentiation ability of human-iPSCs provides an opportunity to use this disease model for understanding the physiology of affected cell types in tissue culture plates. Additionally, this model can be used to screen and identify disease-specific drugs in vitro in a Petri dish. These pre-clinical studies on Petri dishes have provided the first proof of concept and a feasible option for understanding the molecular mechanism of diseases and screening potential molecules for drug development and cytotoxicity studies. The platform for large-scale in vitro drug screening of chemical libraries was called the “disease to Petri dish model” [2,3].

Interestingly, the drug candidates entering into human clinical trials are only those drugs that were screened and pre-tested in pre-clinical research. Ideally, these candidates must work in human clinical trials due to their strict assessment criteria in pre-clinical research. However, a vast inconsistency has been observed during human clinical trials versus pre-clinical research. For example, despite high funding opportunities for clinical trials, up to USD 42.5 billion, the outcomes have been negative, with a 95% failure rate in AD. Moreover, only six drugs indicated for AD were approved by the US Food and Drug Administration (FDA) between 1995 and 2021 [4].

In this review, we discuss the contribution of human iPSCs in scientific and clinical research. To understand the role of human iPSCs in scientific research, we have examined and ranked the most common neurodegenerative diseases, such as PD and AD, diabetic neuropathy, stroke, and spinal cord injury; the results are based on published literature on PUBMED. 

We also explore recent iPSCs-related advances in the field of clinical research and discuss the role of iPSCs in cellular therapy, personalized medicine, and ongoing clinical trials on PD and AD.

## 2. Brief History of iPSCs

The existence of stem cells came into the picture in 1961 when Till et al. described the ability of mouse bone marrow to grow and differentiate into a variety of cell types, later termed pluripotent stem cells [5]. Sir John Gurdon, in 1962, demonstrated for the first time the cellular reprogramming of enucleated unfertilized frog egg cells that were transplanted with the nucleus from epithelial somatic cells of tadpole’s intestine. This reprogramming method was termed somatic cell nuclear transfer (SCNT) [6] and led to the birth of cloning, followed by the somatic cloning of Dolly the sheep in 1997 at the Roslin Institute of the University of Edinburgh in Scotland [7]. This was a breakthrough in the history of stem cell research. This scientific progress in the field of cloning proved that a whole organism could be generated by differentiated somatic cells through egg cells that contain all the necessary factors for reprogramming. In 1998, James et al. isolated human embryonic stem cells for the first time from human blastocysts [8].

The two major discoveries (i.e., the generation of mouse embryonic stem cells (ESCs) cell lines in 1981 and the generation of human ESCs in 1998) have thrown light on ESC capabilities to develop the pluripotent state in any somatic cells of the body [8,9]. In 2001, Tada et al. demonstrated that reprogramming could also be achieved by cell fusion of somatic cells with embryonic stem cells. The fused cell was capable of expressing a pluripotent state [10]. The ESC cell line can be developed from pre-implanted embryos. The above-mentioned research on ESC has provided enormous information for selecting ideal culture conditions and transcription factors for the maintenance of the pluripotent state of the cells.

Developing the pluripotent state in a patient’s specific cells requires reprogramming somatic cells. The three most common different approaches to reprogramming are (1) somatic cell nuclear transfer (SCNT), (2) cell fusion, and (3) direct reprogramming by transcription factors.

SCNT involves the transfer of the nucleus of a somatic cell into an oocyte or early embryo from which the chromosomes have been removed. The somatic nucleus is injected into cloned embryos from mice and humans. This fusion of embryos with somatic cells develops iPSC, where unknown factors of the recipient oocyte are responsible for reprogramming somatic cells into a pluripotent state. 

A paradigm shift has occurred in the last decade after the discovery of human iPSCs reprogramming technology. In 2006, Takahashi and Yamanaka demonstrated for the first time in mice the reprogramming of fibroblast cells into iPSCs by retrovirus-mediated transfection. They further investigated the phenomena and selected 24 pluripotency transcription factors for the study. 

They observed that over-expression of only four reprogramming factors, such as Oct4, Sox2, Klf4, and c-Myc had a vital role and were sufficient to create iPSCs in mouse fibroblasts [11] and human fibroblast to iPSCs [12]. These transcription factors were later called the OSKM factor or Yamanaka factor. Soon after this remarkable discovery, human iPSCs were developed successfully from human fibroblasts in late 2007 by Yamanaka’s and Thomson’s groups using a similar approach and with different types of reprogramming factors such as (OCT4, SOX2, NANOG, and LIN28) [12,13]. In 2008, Nakagawa et al. observed that iPSCs can be generated from fibroblasts using only three reprogramming factors, such as Oct4/Klf4/Sox2 without c-Myc [14].

Human iPSCs can also be developed from patients’ somatic cells and applied for biomedical research for developing disease-specific in vitro models for drug discovery and development [2]. Additionally, human iPSCs can also be used to develop personalized and precision medicine [15,16]. 

Since the discovery by Dr. Shinya Yamanaka in 2007, several new methods have been optimized to improve the induction efficiency of iPSCs and advanced reprogramming methods. The use of chemical compounds and growth factors has improved the induction efficiency of iPSCs. In 2008, Shi et al. demonstrated that mouse embryonic fibroblasts could be induced into iPSCs using small molecules through reprogramming of Oct4/Klf4 and can compensate for Sox2 [17]. In 2008, Huangfu et al. observed that small molecules, such as 5-azacytidine, valproic acid, histone deacetylase inhibitor, and DNA methyltransferase inhibitor, can improve the reprogramming by 100-fold. Moreover, primary human fibroblasts can also be reprogrammed using valproic acid through efficient programming of Oct4 and Sox2 only [18,19]. This ground-breaking discovery facilitated the use of stem cell technology and found a way to overcome the long-lasting ethical controversies associated with hESCs or nt-hESCs research. Moreover, the simple and easy process of viral transduction to generate human iPSCs made the platform ideal for iPSCs generation.

To date, different types of stem cells have been used in stem cell research, such as ESCs, very small embryonic-like stem cells (VSELs), nuclear transfer stem cells (NTSCs), reprogrammed stem cells (RSCs), and adult stem cells (ASCs). Interestingly, ESCs, iPSCs, and ASCs are used to generate tissues and organs in vitro for developing treatments. However, only NTSCs have shown potential to develop cloned animals such as sheep [7], mice [20], cattle [21], pigs [22], cats [23], rats [24], and dogs [25]. In total, 23 cloned mammalian species have been developed [26].

## 3. iPSCs in Scientific Research

To understand the role of iPSCs in scientific research, we thoroughly investigated the published literature in the PUBMED MEDLINE database using specific keywords. However, data acquired from PUBMED are based on an algorithm and solely depend on the mapping of specific keywords within the published literature, such as articles/reviews/clinical trials. Still, we can observe the trend and significance of iPSC-related research and its contribution to science. A similar analysis was published by the author to select the most popular diabetic animal models in 2019 [27] and to understand the contribution of peptides in diagnostics in 2021 [28]. The database was searched using the primary keyword “iPSC” with two additional filters, “Human” and “Animal”. The output of the result shows that there has been a constant increase in the number of publications related to iPSCs research since 2007. As per the published literature in PUBMED, we also observed that human-related iPSCs research has been growing considerably compared to animal-related iPSCs research since 2011. In 2018, human-related iPSCs publications reached 2033, which was 2 times higher than animal-related iPSCs publications (Figure 1). This clearly shows the significant role of iPSCs in human-related biomedical research. 

Furthermore, the contribution of iPSCs-related research in most common diseases was also evaluated and categorized as per their publication in the last decade (w.e.f. 1 January 2012 to 31 December 2021) in the PUBMED MEDLINE database. The database was searched by using primary keywords “iPSC”, and an additional nine filters for different diseases were used to discriminate the data (i.e., “Cancer”, “Heart disease”, “Neurodegenerative disease”, “Diabetes”, “Liver disease”, “Autoimmune disease”, “Cartilage regeneration”, “Zika virus”, “SARS-CoV-2”). In our customized search, we observed that most of the published literature in the last decade was on cancer, which was 3155, followed by heart disease, 2251, and neurodegenerative disease, 1609. In conclusion, cancer-related iPSCs research ranked first, and heart-related iPSCs research ranked second, followed by neurodegenerative-related iPSCs research which ranked third in the last decade (Figure 2).

To understand the role of iPSCs research in neurodegenerative diseases, we further examined the published literature on neurodegenerative diseases in PUBMED. The database was searched using the primary keyword “iPSC” along with eight additional filters for common neurodegenerative diseases such as “Spinocerebellar Atrophy”, “Multiple Sclerosis”, “Ataxia”, “Spinal Cord Injury”, “Huntington’s Disease”, “Amyotrophic Lateral Sclerosis”, “Alzheimer”, and “Parkinson”. 

In our analysis, we observed that the contribution of iPSC-related research for Parkinson’s was highest, with 932 published literature, followed by Alzheimer’s with 761 published literature on PUBMED (Figure 3).

## 4. Role of iPSCs in Neurodegenerative Diseases

There is an unmet need for clinicians to find new therapies for neurodegenerative diseases. Despite good clinical research, it has been acknowledged that discovered drug therapies are not up to the mark. To accelerate the drug discovery pipeline for neurological diseases, several animal disease models have been developed in the past [29,30,31]. Unfortunately, the lack of equivalent animal models has minimized the success rate of new therapies and caused poor initial screening and assessment of the molecules used in pre-clinical research. More than 90% of pre-clinically successful drugs fail during the final step of the human clinical trial. This suggests that animal models are often poor predictors of human biology [32]. Creating a disease model of degenerative diseases is a difficult task. Research scientists have either dysregulated the gene expression of a specific gene to develop a cell culture model or created a knockout animal model. However, these models could not be considered an ideal disease model which resembles human pathology. Due to ethical issues, it is also very difficult to obtain human brains postmortem to conduct scientific research. Even if a human brain is provided, brain tissues are highly degradable and immunologically mature for research. 

In the last two decades, it has become possible to establish pluripotent stem cells from the somatic cells of any individual without knowing his race and genetic background. Therefore, human iPSC-derived cell cultures became a unique platform to study ex vivo phenomena, particularly for nervous system disorders. 

### 4.1. iPSCs in PD

PD is associated with mutations in different genes such as SNCA, LRRK2, VPS35, Parkin, PINK1, and DJ-1 [33]. The CHCHD2 mutation is also associated with PD [34]. Human iPSCs line from PD patients were successfully developed. Wang et al., 2018 developed iPSCs from dermal fibroblasts of 52-year-old PD patients by transfecting the cells with episomal plasmids expressing OCT3/4, SOX2, KLF4, LIN28, and L-MYC. The developed iPSCs line (ZZUi007-A) harbors a CHCHD2 mutation [35]. Takahashi et al., 2007 developed an iPSCs line (201B7) from the dermal fibroblasts of a healthy donor. Fibroblasts were reprogrammed using retroviral transduction expressing OCT4, SOX2, KLF4, and MYC at Kyoto University [12]. They serve as a “normal” control [36]. Imaizumi et al., 2012 generated human iPSCs from two familial forms of PD patients with mutations in the parkin gene. Enhanced activity of the nuclear factor erythroid 2-related factor 2 (Nrf2) pathway, along with increased oxidative stress, was observed in human iPSC-derived neurons with PARK2 mutations [37]. Suda et al., 2018 generated a human iPSCs line (B7PA21) from PD patients with PARK2 mutations. It was observed that the expression of ghrelin receptors in PD-iPSC-derived dopaminergic neurons was down-regulated compared to healthy controls. Moreover, generating the PARK2-KIKO line from 201B7 through CRISPR Cas9 technology also mimicked the loss of function of the PARK2 gene [38]. Shiba-Fukushima et al., 2017 demonstrated that phospho–ubiquitin signaling was affected in human dopaminergic neurons containing Parkin or PINK1 mutations. It was also observed that regulation of axonal mitochondrial transport and phospho–ubiquitin signaling was compromised in human dopaminergic neurons containing Parkin or PINK1 mutations [39]. The importance of voltage-gated calcium channel in neurodegenerative diseases, especially in PD, has been confirmed. Few studies related to human iPSCs have described a potential link between Ca2+ oscillations and the susceptibility of the dopaminergic neurons to neurodegeneration. Cav2.3 knockout mouse showed upregulation of NCS-1, a Ca2+-binding protein involved in neuroprotection. The data were supported by human iPSCs line from PD. Similarly, the NCS-1 knockout mouse exacerbated nigral neurodegeneration and downregulated Cav2.3. [40]. Grigor’eva et al., 2021 demonstrated that the iPSCs line (ICGi034-A) could be obtained from PBMCs of PD patients with heterozygous c.1226A > G (p.N370S) mutation by reprogramming to study the pathogenesis of GBA-associated PD [41]. Schweitzer et al., 2020 showed the implantation of iPSCs dopaminergic progenitor cells to the midbrain of PD patients via autologous transplantation. Clinical grade iPSCs were generated in vitro and were first tested on a humanized mouse model to check the immunogenicity and then implanted into the putamen of PD patients without any immunosuppression. Positron emission tomography with the use of fluorine-18-L-dihydroxyphenylalanine suggested graft survival [42].

Chen et al., 2021 demonstrated that an iPSCs line could be developed from PBMCs of a 32-year-old PD patient with homozygous mutation of c.189dupA in PARK7 (FJMUUHi001) by reprogramming five factors, OCT3/4, SOX2, c-MYC, KLF4, and BCL-XL. The induced iPSCs could be differentiated into three germ layers and were able to express the markers of pluripotency [43]. Moreover, all the cell lines developed for PD can be seen in Appendix A.

### 4.2. iPSCs in AD

Michael Peitz et al., 2018, demonstrated that peripheral blood cells of a male AD patient could be developed into a human iPSCs line by employing Sendai virus vectors which encode for OCT4, SOX2, KLF4, and c-MYC transcription factors [44]. The characterized iPSCs line expresses differentiation into all three germ layers and also maintains APOEε4/ε4 allele, a prominent risk factor for sporadic late-onset AD [45,46]. Zhang et al., 2021 demonstrated the reprogramming of PBMC by using a non-integrating episomal vector system. The reprogramming was achieved by retroviral transduction OCT3/4, KLF4, SOX2, and c-MYC using the Sendai virus. A human iPSCs line SIAISi004-A from PBMCs of a 74-year-old AD female was developed [47]. Liu et al., 2020 developed the iPSCs from sporadic Alzheimer’s disease (sAD) patients. The reprogramming was done on PBMCs using the Sendai virus expressing Oct3/4, Sox2, c-Myc, and Klf4 transcription factors [48]. Cusulin et al., 2019 used human iPSCs derived from sporadic AD patients. Developed iPSC neural cell lines were used for the characterization of secretase modulators compounds that reduce the production of Aβ42 [49]. Takayuki et al., 2017 used AD-derived human iPSCs for screening of anti-Aβ cocktail mixture from the pharmaceutical compound library of 1258 compounds. A mix of topiramate, cromolyn, and bromocriptine was identified, which was able to reduce Aβ deposition and plaque formation [50]. Chang et al., 2019 developed iPSCs from familial AD patients with heterozygous D678H mutation in the APP gene. AD-derived human iPSCs have been used to screen the compound having the ability to reduce Aβ aggregation and improve neuronal viability and neurite outgrowth. The identified compound was indole compound NC009-1 (3-((1H-Indole-3-yl) methyl)-4-(2-nitrophenyl) but-3-en-2-one) [51]. Wang et al., 2021 demonstrated that iPSCs could be derived from peripheral blood mononuclear cells of a 70-year-old male donor with APOE-ε4/ε4 alleles. The induced iPSCs were able to express pluripotency markers and showed normal karyotypes with differentiation potential [52]. Zhang et al., 2021 generated iPSCs from peripheral blood mononuclear cells of an 87-year-old female donor with APOE3 (ε3/ε3) alleles. More than 97% iPSCs were able to express pluripotency markers such as NANOG, OCT4, and SSEA4 along with normal karyotype. The developed iPSCs line could be considered one of the valuable resources for studying sAD pathogenesis in vitro [53]. Moreover, all the cell lines developed for AD can be seen in Appendix A.

### 4.3. iPSCs in Diabetic Neuropathy

Diabetic neuropathy (DN) is the most common and earliest complication of diabetes mellitus (DM), which is often diagnosed in the progressive state [54]. It affects 50% of patients with DM and about 10–25% of prediabetic patients [55]. The pathogenesis of DN is complex, including, primarily, hyperglycemia, which is associated with biochemical processes leading to the overproduction of reactive oxygen species, increased expression of pro-inflammatory cytokines, and abnormal levels of gas transmitters [56]. These processes affect the function of different cell types, including neurons and Schwann cells but also endothelial cells, leading to the reduction of nerve blood flow and degeneration of nerve fibers with consequent dysfunction of peripheral nerves [57]. Different pathological changes, including axonal loss and/or degeneration, as well as demyelination, have been observed in diabetic organisms. Schwann cells play a pivotal role in the appropriate functioning of peripheral axons, while they cover and support myelinated as well as non-myelinated fibers of the peripheral nervous system. This support includes not only physical but also chemical mechanisms, while Schwann cells release several neurotrophic substances [58]. Based on this information, the effect of Schwann cell application in animal models of peripheral neuropathy was tested by different research groups. Himeno et al. demonstrated that some mesenchymal stem cells, (MSC)-like cells derived from iPSCs, if transplanted to diabetic mice thigh, engraft to the peripheral nerve and express S100β, a Schwann cell marker. This observation indicates the ability of grafted cells to directly construct peripheral nervous tissue. Additionally, transplantation of MSC-like cells exerted a beneficial effect on blood flow and capillary number in the soleus muscle of diabetic mice. From a functional point of view, treatment with MCS-like cells ameliorated physiological impairments caused by diabetes, demonstrating the beneficial effect of such treatment on diabetic peripheral neuropathy [59]. The positive therapeutic effect of Schwann cell transplantation was demonstrated in injury-induced peripheral neuropathy 30 years ago in rats [60] and later on also in humans [61,62]. The source of Schwann cells for transplantation was both the sural nerve and damaged sciatic nerve stump in these cases. Another source of autologous Schwann cells may be the patient’s skin. The Schwann cells thus obtained exhibit the same properties as Schwann cells obtained from the nerve [63]. Unfortunately, a similar comparison between human skin-derived Schwann cells and iPSCs is not yet available.

In the past, a number of studies have been conducted that have focused on the use of stem cells, especially MSCs and bone marrow mesenchymal stem cells (BMMSCs), in the treatment of diabetic peripheral neuropathy. The results of many of them indicated a positive effect of such treatment based on in vitro or animal experiments. A diabetic environment causes damage to endogenous stem cells, which leads to severe diabetic complications [56]. Replacing these damaged cells with exogenous cells could be the right way to go. Unfortunately, a clinical study demonstrating the safety and efficacy of such therapy is not yet available [56]. 

To our knowledge, to date, no clinical trial involving using iPSCs as a therapeutic tool in patients with diabetic neuropathy has occurred. However, there is a very interesting study where iPSCs obtained from a patient with idiopathic small fiber neuropathy were used to determine the optimal treatment for that patient [64]. This case points to the significant possibilities of using iPSCs in the development and testing of drugs in general and their use in personalized medicine in the search for optimal treatment of diabetic neuropathy or other diseases. 

### 4.4. iPSCs in Stroke

Stroke is the second leading cause of death globally [65]. In the majority of stroke cases, cerebral blood supply is decreased due to thrombotic and/or embolic processes leading to ischemia of nervous tissue with subsequent cell apoptosis and necrosis. The extent of a patient’s neurological impairment depends on the location and size of the ischemic lesion. Currently, therapeutic approaches are based primarily on the effort to reopen a closed vessel using thrombolytic agents or mechanical removal of the thrombus. The final effect of the performed therapy depends very much on the time lapse between the first symptoms and the implementation of the therapy. If the recommended therapy is used later, i.e., outside the therapy window, the patient’s condition may worsen due to the leaching of accumulated ROS and subsequent neuroinflammation [65]. From the above, it is clear how important it is to find new, more effective therapeutic approaches. Due to the fact that necrosis of nerve tissue cells occurs during the ischemic process, it is proposed to solve this situation by transplanting the cells into the damaged area. Researchers have been studying this possibility for decades using various stem cells, such as ESCs, NSCs, or MSCs, but in recent years also iPSCs.

Up to now, a variety of animal experiments have been conducted in order to evaluate the effect of iPSCs application after stroke. They demonstrated that iPSCs could cause a reduction in lesion volume, improvement of sensorimotor functions, and promotion of neurogenetic and angiogenetic processes [66]. In addition, they exert immunomodulatory and anti-inflammatory effects [67]. Results of iPSCs transplantation experiments up to 2020 are summarized in a nice review written by Duan and coauthors [66].

A recently published study demonstrated that transplantation of iPSCs into rat brains affected by stroke causes a positive effect on glucose metabolism restoration in the ischemic area and attenuation of neurofunctional deficits. These effects were more prominent than those obtained by transplantation of NSCs [68]. Transplanted iPSCs are able to differentiate into nerve cells and angiogenic cells and cause changes in the microenvironment, including different expressions of multiple proteins related to oxidative stress, mitochondrial function, axonal remodeling, and neuronal survival [68].

Nerve cells and pericytes, cells involved in forming the blood–brain barrier (BBB), are both involved in the pathophysiology of stroke [69]. Pericytes generated from human iPSCs were transplanted to a mouse model of stroke, where they helped to reconstruct the BBB and promote functional recovery [70]. Unfortunately, the effect of pericytes in stroke recovery processes could also be adverse; they could contribute to scar formation and inhibit axonal regeneration [69]. 

Recently, researchers have been focusing on the paracrine concept of cell therapy, where conditioned culture medium could serve as a therapeutic tool because it contains biologically active molecules. Salikhova and coworkers tested the effect of the application of human iPSCs-derived glial and neuronal progenitor cells-conditioned medium to a rat model of stroke. Intra-arterial administration of these media caused diverse effects. Application of glial progenitor cells-conditioned medium led to a decrease in neurological deficit, affected the expression of pro- and anti-inflammatory cytokine genes, and improved re-vascularization of the damaged area, while such effects were not observed after application of neuronal progenitor cell-conditioned medium [71]. This is most likely caused by the different compositions of these media and suggests the importance of glia in the whole regenerative process after a stroke. Applying the culture medium instead of transplanting the cells may be a way to eliminate the various risks associated with transplanting iPSCs in the patient.

### 4.5. iPSCs in SCI

Within the central nervous system (CNS), trauma could affect the brain, that is, traumatic brain injury (TBI) or spinal cord causing SCI. Damage to the nervous tissue of the CNS finally leads to progressive and massive neuronal cell loss and axonal degeneration. Cell death occurs in the acute phase as a result of traumatic force and subsequently in the subacute phase through acute inflammatory processes and/or ischemia. The subacute phase lasts from the 2nd to 14th day after the trauma, and during this period, there is more excessive degeneration of tissue than in the acute phase [72]. This fact is used in determining the strategy of therapy and also indicates the potential effect of therapy based on cell transplantation, where transplanted cells could replace lost cells and also release growth factors improving the inner environment of damaged nervous tissue, protecting host cells as well as its own cells [72]. Over the past thirty years, this topic has been studied by many scientists who focused on the effect of different types of stem cells, including mesenchymal stem cells, neural stem cells, neural progenitor cells, embryonic stem cells, and iPSCs [73]. 

Neural progenitor cells (NPCs) derived from iPSCs exert characteristics different from those of NPCs derived from fetal cells, e.g., in gene expression patterns and epigenetic status. Nowadays, techniques allowing the preparation of region-specific NPCs from iPSCs are available [74]. They are based on using different gradients of morphogens during neural induction of iPSCs [75,76]. So far, these possibilities have been used mainly for modeling various CNS diseases and searching for optimal drug therapies [77]. Neurodegenerative diseases are very complex conditions; their pathophysiology involves not only neurons but also supporting nerve cells, the most abundant of which are astrocytes. Astrocytes produced from iPSCs are also used in disease modeling and drug screening. However, the results of a recent study suggest that astrocytes may be a source of immunoglobulins in trauma-damaged neural tissue [78]. What role this fact plays in the process of nerve tissue regeneration after trauma will have to be determined in further experiments, where astrocytes obtained from iPSCs could also be used.

Up to now, a big effort has been made to study the effect of iPSCs-derived NPCs application on the regeneration of nervous tissue after SCI. These cells are able to survive and differentiate at the site of injury. Additionally, they reduce the level of pro-inflammatory cytokines after SCI, thus reducing the formation of glial and fibrotic scars [79]. Many studies have demonstrated functional recovery of injury after their transplantation [80], suggesting their potential in SCI treatment. 

## 5. Role of iPSCs in Personalized Medicine 

A significant role of iPSCs in cellular therapy has been observed, which may lead to human clinical trials and provide a future roadmap for therapy. Additionally, patient-derived iPSCs can act as a unique model for understanding disease development. It can also help in drug screening and provide new insights for developing ‘new future medicine’. Regeneration therapy has emerged as a newly developed field of personalized medicine based on cellular therapy. Personalized medicine is developing a specific medication for an individual based on the individual’s pharmacogenetics and pharmacogenomic information [81]. 

It is important to study personalized medicine or personalized pharmacology because every person has individual heterogeneity to different types of diseases. It could be due to multiple causative factors such as genetics, epigenetics, environment, or demographic, including age, sex, and ethnicity. These factors together can drive the progression of any disease. However, some authors have claimed that genetic factors are the strongest risk factors in complex diseases, including neurological disorders. Furthermore, crosstalk between genetic, environmental, demographic, and lifestyle factors plays a crucial role in disease development. On the contrary, there are few examples where individual heterogeneity does not follow the standard Mendelian patterns of inheritance. 

The discovery of iPSCs technology in 2007 [12] revolutionized the field of personalized medicine by providing additional ways of drug screening and can also be considered an appropriate candidate for personalized cell therapies [81,82]. The main objective of regeneration medicine is to repair cells or tissues of any organs which are damaged due to aging, chronic diseases, neurological diseases, congenital, or any other abnormalities, etc. The different types of stem cells are used in regenerative medicine, which is pluripotent in nature. Among all types of stem cells, iPSCs are considered to be the most appropriate cells for therapies. Isolated stem cells are edited via gene editing and used for cellular therapy. For transferring desirable genes, two commonly used methods are in vivo and ex vivo. In in vivo therapy, new genes are directly introduced via a plasmid or viral vectors or clustered regularly interspaced short palindromic repeats (CRISPR) strategy [83,84]. The limitations of in vivo gene therapy are associated with gene silencing, mutations, non-specific gene expression, and immune reaction against the vector [85]. In ex vivo gene therapy, cells are first modified in vitro and characterized before the transplant so that patients are not directly exposed to vectors and provide safe, stable, and transient grafts [86]. 

The recent development in the field of stem cell research has advanced the cutting-edge of ex vivo gene therapy by generating human iPSCs from the patient’s blood or skin [87,88]. For example, patient-specific iPSCs allow us to research a wide range of incurable disorders, such as neurodegenerative diseases of the CNS, heart infarction, diabetes mellitus, and liver, lung, and kidney disease. 

To explore the therapeutic role of iPSCs in neurodegenerative diseases, we carefully examined the database of the International Clinical Trials Registry Platform (ICTRP), WHO. The registered clinical trials for neurological disorders were analyzed on https://clinicaltrials.gov/. The data was accessed on 18 February 2022 by using keywords for the disease condition “Neurologic Disorder” along with one additional filter for “iPSC”. We found that 30 clinical trials were registered. Out of 30 trials, 2 trials were terminated, 1 trial was suspended, 1 was withdrawn, and 3 trials showed their status as unknown, meaning they had passed their completion date with no updates for more than 2 years. Only seven clinical trials were completed. Out of these seven trials, we observed that only four clinical trials “NCT02980302”, “NCT03883750”, “NCT03867526”, and “NCT01639391”, were designed with the main objective of generating patient-derived iPSCs (Appendix A). However, the other three clinical trials were designed with a different aim and had the secondary objective of retaining biospecimen for iPSCs line generation and bio-repository. Moreover, we did not find any completed clinical trials on iPSCs for “Alzheimer” or “Parkinson”; most of the trials were designed for ALS. To confirm our results, two separate searches were performed using keywords for the disease conditions “Alzheimer” and “Parkinson”. We found that only one clinical trial was registered for “Alzheimer”, i.e., NCT00874783, and two for “Parkinson”, i.e., NCT00874783, NCT01143454.

Interestingly, we observed that most of the trials for disease conditions “AD” and “PD” were designed for “Stem Cell Therapy” and “Gene Therapy”. To understand the current scenario of clinical trials for PD and AD, we analyzed the database for the last decade (w.e.f. 1 January 2012 to 31 December 2021). A total of four separate searches were made for “Stem Cell Therapy” with disease conditions such as “Neurologic Disorder”, “Neurodegenerative Disorder”, “Alzheimer”, and “Parkinson”. We observed that the total number of clinical trials registered for neurologic disorders was 381, out of which 97 trials were for neurodegenerative disorders and 44 trials were for AD and PD.

As mentioned above, we used a similar strategy to obtain the clinical trial data for “Gene Therapy”. We have observed that the total number of clinical trials registered for neurologic disorders was 398, out of which 100 trials were for neurodegenerative disorders, and 25 trials were for AD and PD. Data showed that 25% of clinical trials registered for neurodegenerative disorders belong to AD and PD.

In conclusion, we found that clinical trials for “Stem Cell Therapy” have been growing very rapidly, and 45.3% of clinical trials registered for neurodegenerative disorders belong to AD and PD in the last decade. However, only 25% of registered clinical trials for “Gene Therapy” for neurodegenerative disorders belong to AD and PD (Figure 4).

### 5.1. Are Clinical-Grade Allogeneic iPSCs Important?

The first successful autologous transplantation of iPSC-derived retinal pigment epithelial cells (RPEs) into a human was accomplished in 2014 by Masayo Takahashi’s team at the RIKEN Center for Developmental Biology, Japan. Before surgery, iPSCs were developed from two patients’ skin fibroblasts. One patient was a 77-year-old woman and the other was a 68-year-old man; both were diagnosed with polypoidal choroidal vasculopathy. Therapy was provided after testing the tumorigenic potential of Patient1-28-RPE cells in immunodeficient mice (nonobese diabetic/Shi-scid/IL2rγnull [NOG] mice) in only one patient (77-year-old woman). In the follow-up study, no serious adverse event was noticed [89]. The above-mentioned study on iPSCs showed the potential of iPSCs to be used for autologous transplantation. Furthermore, many clinical trials have been registered and initiated. Most of them are in Phase I/IIa stages; details are given in Appendix A. Data suggest that only three patients worldwide have received iPSC-derived autologous transplantation therapy, and none have suffered from serious adverse events [90].

iPSC-based therapies can be developed for a wide range of disorders and become a universal therapy but two important factors, such as cost and time, are high. Presently, the total cost for developing autologous iPSC-based therapy is approximately USD 1 million and takes approximately 6 months. It includes a number of steps and quality checks before the final transplantation of iPSCs to patients. 

To minimize the cost and time of developing iPSC-based therapies, it has been advised to use allogeneic iPSCs from “super donors”. Super donors are healthy people who are homozygous at the major HLA loci. In 2013, the Center for iPS Cell Research and Application (CiRA), Kyoto University, started to develop clinical-grade allogeneic iPSCs from super donors. This facility is also used for the distribution of iPSCs to other centers for regenerative therapy in Japan. The three most common major HLA haplotypes are HLA-A, HLA-B, and HLA-DRB1. The use of these three allogeneic iPSCs from super donors has increased the probability of donor–recipient matching and serves approximately 30% of the Japanese population. However, the target is to cover at least 50% of the Japanese population in the future [91].

### 5.2. iPSCs in Personalized Pharmacology

A new avenue in scientific research has advanced in vitro models such as “organ-on-a-chip” (OoC) technology. The concept of OoC technology is a revolutionary way of screening drugs suitable for human clinical trials. Perestrelo et al., 2015 reviewed interesting advancements in the field of microfluidic-based devices and their applications in biomedical fields, such as the body-on-a-chip concept [92]. 

The OoC technology has great potential to mimic the natural physiological environment and functions of human organs. An interdisciplinary approach has been developed to integrate 3D cell culture conditions on microfluidics devices for in vitro analysis of drug screening. It has revolutionized the field of drug screening and toxicology studies. It has also been observed that proliferation, migration, differentiation, drug toxicity resistance, and gene expression could be impacted significantly under different culture conditions, such as 2D versus 3D [93]. 

The reprogrammed iPSCs derived from patients with different genetic backgrounds can also be used to analyze the efficacy and safety of drugs in personalized medicine with precise genetics of the individuals wherein iPSCs can be grown on 3D matrices inside the microfluidic devices using techniques such as micromachining, 3D printing, and hydrogels. These conditions represent a unique approach closer to in vivo conditions. 

The OoC technology is used to study the potential impact of any drug, such as ADME (absorption, distribution, metabolism, and excretion) and other toxicities. These models include gut-on-a-chip [94], liver-on-a-chip [95,96], kidney-on-a-chip [97] and heart-on-a-chip [98,99,100,101], blood-brain-barrier-on-a-chip [102,103], and brain-on-a-chip [104]. 

To study the absorption of any drug, patients’ iPSC-derived intestinal organoids micro-engineered chips were used, which mimic inflammatory bowel disease [94]. 

The most common cause of drug failure is drug-induced hepatotoxicity. Recently, iPSCs-derived hepatocytes or iPSCs-derived liver organoids were used to study drug metabolism, detoxification, and hepatotoxicity on a chip. For example, drugs such as Terfenadine, Tolcapone, Trovafloxacin, Troglitazone, Rosiglitazone, Pioglitazone, Lipopolysaccharide (LPS), and Caffeine were examined on a microfluidic platform with a four-cell liver acinus microphysiology system including PHH or iPSC-HEPs and three different human cell lines for NPCs. The assay was performed for nine days and exhibited upregulation of mRNA level of the drug-metabolizing genes along with the increased activity of CYP450 under 3D cell culture compared to monolayered 2D static conditions [105]. 

Fanizza et al., 2022 reviewed and elaborated on the role of iPSCs in the screening of drugs for personalized medicine, especially for neurodegenerative diseases. The translational value of OoC was analyzed to create more realistic disease models. The value of OoC has been tremendously enhanced by using patient-specific iPSCs for developing a new generation device called “patient-on-a-chip” [106]. 

Furthermore, multi-OoC devices could also allow the crosstalk between varieties of cells of different origins that mimic a natural physiological condition very close to in vivo conditions and could be useful to study the pharmacodynamic and pharmacokinetics of drugs in personalized medicine.

## 6. Limitations and Challenges

The recent development in iPSCs technology has opened a unique avenue for clinical research. However, obstructions such as irreproducibility, epigenetic variations, genetic instability, high cost, and time are a concern for clinicians and researchers (Figure 5). Subsequently, researchers have raised safety concerns about using iPSCs in patients because of their vulnerability to genetic variations, tumorigenicity, which may cause loss of immunogenicity, and graft rejection in a few cases. Mutations can occur at any time during the reprogramming process or during the maturation process when cells are cultured in vitro many times. However, recent studies on iPSCs-based cell therapy have demonstrated that autologous cell transplantation could be obtained without using immunosuppressants or without rejection of the allograft. 

New pathways generate a new era of responsibilities. A unique opportunity for the pharmaceutical industry and clinical research has arisen with the advancement in stem cell research. New laws and standards must be approved for the use of iPSCs. However, unnecessary barriers hindering iPSCs-related research must be curtailed. Due to private information present in the form of DNA, donors’ information must be protected. Some basic approval must be obtained to ensure the ethical integrity of iPSCs production and its application. The donor must be informed about the project timelines and research topic so that they are aware of the use of their cells with time duration. Informed consent must be obtained for all the participants, including donors and recipients. To ensure quality and consistency, iPSCs should be processed in a controlled environment according to quality standards. It would be best to carry out iPSC-related research under GMP conditions enforced by the US FDA under the authority of the Federal Food, Drug, and Cosmetic Act. It covers all the aspects of production, starting from somatic cell isolation to generating an iPSCs line and its application. All procedures should be documented in the form of standard operating procedures (SOPs), procedure descriptions, registries, records, training documents, manuals, lists, and others to assure the good management, functionality, and traceability of the cell bank should be used to minimize the variations which will also provide a documented proof and consistency of the process at each step. Baghbaderani et al., 2015 developed master cell banks (MCBs) under cGMPs [107]. In 2020, Rivera also developed iPSCs using mRNA under cGMP conditions [108]. 

Despite the many ethical advantages of iPSCs over ESCs, new ethical–social issues have arisen, such as the use of iPSCs for human cloning, human–animal chimeras, interspecies chimeric animals, and illegal generation of human gametes. Other potential illegitimate usages of iPSCs should also be considered very carefully and are the topics of further discussion. It is also important to understand the ethical issues related to the use of tetraploid complementation technology in humans [109,110,111]. These ethical–social issues can become more complex if we also consider copyright and patent-related issues with iPSCs generation.

## Figures and Tables

**Figure 1 jpm-12-01485-f001:**
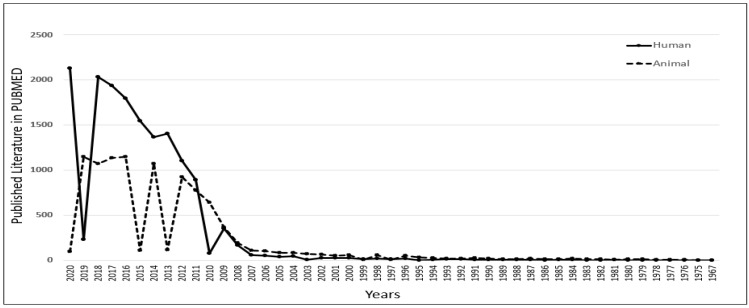
Understanding the role of induced pluripotent stem cells (iPSCs) in scientific research. Published data with the keyword iPSCs were compared in PUBMED for human versus animal. It was observed that there was an increase in human-related iPSCs research from 2011.

**Figure 2 jpm-12-01485-f002:**
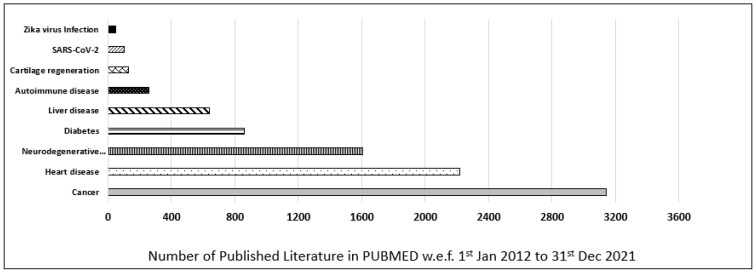
Published data for iPSCs research in the last decade. Comparison of different diseases such as cancer, heart disease, neurodegenerative disease, diabetes, liver disease, autoimmune disease, cartilage regeneration, Zika virus, and SARS-CoV-2 based on scientific research published in PUBMED in the last decade (w.e.f. 1 January 2012 to 31 December 2021).

**Figure 3 jpm-12-01485-f003:**
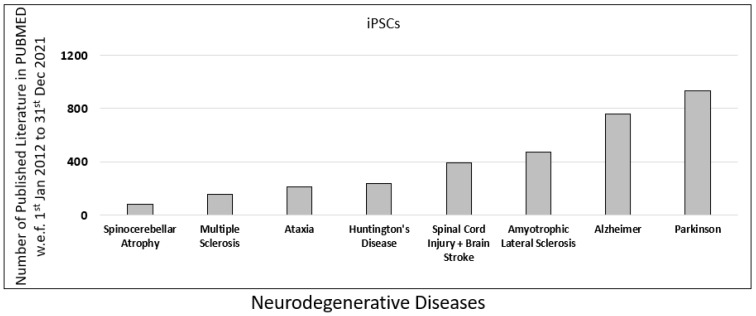
Published data for iPSCs research in the last decade. Comparison of different neurodegenerative diseases such as spinocerebellar atrophy, multiple sclerosis, ataxia, spinal cord injury (SPI), Huntington’s disease, amyotrophic lateral sclerosis (ALS), Alzheimer’s disease (AD), and Parkinson’s disease (PD), based on scientific research published in PUBMED in the last decade (w.e.f. 1 January 2012 to 31 December 2021).

**Figure 4 jpm-12-01485-f004:**
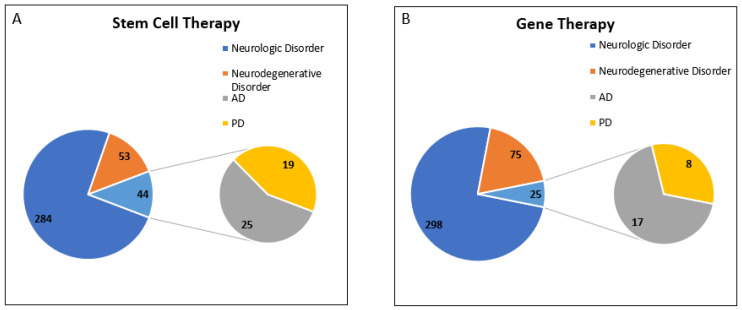
Registered Clinical trial for PD and AD in last decade (w.e.f. 1 January 2012 to 31 December 2021). Analysis of clinical trial data obtained from https://clinicaltrials.gov/ was conducted. Data was accessed on 18 February 2022 (**A**) Total number of clinical trials registered for neurologic disorders for stem cell therapy was 381, out of which 97 trials were for neurodegenerative disorders, including 44 clinical trials for AD and PD. (**B**) Total number of clinical trials registered for neurologic disorders for gene therapy was 398, out of which 100 trials were for neurodegenerative disorders, including 25 clinical trials for AD and PD.

**Figure 5 jpm-12-01485-f005:**
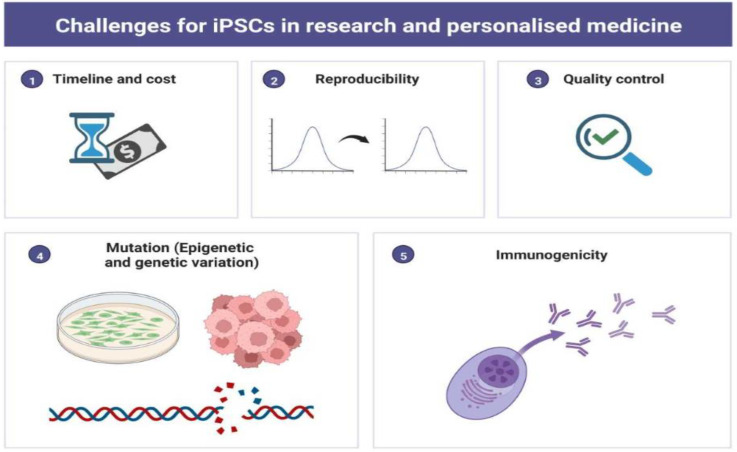
Pictorial Diagram Representing Challenges for iPSCs in Scientific Research and Personalized Medicine. The figure has been created with BioRender.com, accessed on 2 September 2022.

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
