# Peer review of "iPSCs in Neurodegenerative Disorders: A Unique Platform for Clinical Research and Personalized Medicine"

_jpm, 2022, doi:10.3390/jpm12091485_

Round 1

Reviewer 1 Report

Dear Editor

I reviewed the above mentioned review article, in this paper, the authors represent the iPSCs technology in neurodegenerative disorders such as Parkinson's disease and Alzheimer's disease. At the first, the authors introduced the “brief history of iPSCs” technology and “iPSCs in scientific research”. After that, the authors wrote about the iPSCs technology in “Neurodegenerative Disorders”, “Diabetic Neuropathy”, “Stroke”, and “Spinal Cord Injury”.

Although the chosen topic of MS is an interesting one and could help with authorship, the contents of the study do not go far enough in defining its main objectives and some redundant data are included in the study that undermines its main objectives. for instance, in some parts authors inclined to the theraputic benefits of iPSCs. these issues have weaken the novelty of the study.

Additionally, like other scientific fields, this field has several limitations when it comes to extrapolating its results to the clinical field, therefore, authors should mention the limitations and challenges associated with this technology.  Further, the authors are advised to include new figures and tables with their article. finally, it is obvious that acronyms should be introduced in their first appearance in the text. in this MS this issue is not being addressed. For instance, some acronyms are used without being introduce and then in next paragraph they are bring as full term. (please see: iPSCs, AD, PD, ALS, ....)

Reviewer 2 Report

Congratulations to the team for this an amazing effort specially related to iPSCs and Personalized medicine.

Before finalising the manuscript please have a look into the spellings and abbreviations one more time.

1. The ECS cell (line-81) correct it into ESC

2. And write the full name of ESC in starting

Reviewer 3 Report

In the article entitled "iPSCs in Neurodegenerative Disorders: A Unique Platform for Clinical Research and Personalized Medicine" the authors review extensively the use of induced pluripontent stem cells for the treatment of various neurodegenerative diseases.
Although cell reprogramming technology has been advancing significantly, mostly in-vitro, it has not yet yielded significant results in clinical trials.
The authors present a case of two patients diagnosed with polypoidal choroidal vasculopathy, where the results show that one of the patients neither improved nor worsened after transplantation.
The use of this technology will undoubtedly be of benefit to people with neurodegenerative diseases, however, there are still several aspects that need to be taken into account, especially in personalized medicine.
These aspects would be:
1. Why does the person have that disease and not another one?
2. What are the factors that led him/her to acquire the disease?
3. In neurodegenerative diseases it is very important to take into consideration neuroinflammation.
4. The best study model for these diseases is the human, however it is very important to cover the ethical aspects.

The article is very interesting and well described. Undoubtedly, if this technology is perfected, in practice it will be of great benefit to human beings.

Round 2

Reviewer 1 Report

In the revised MS, the authors have made substantial improvements, and it can now be published.